# The Effect of a Structured Individualized Educational Intervention on Breastfeeding Rates in Greek Women

**DOI:** 10.3390/ijerph182111359

**Published:** 2021-10-28

**Authors:** Theoni Truva, George Valasoulis, Abraham Pouliakis, Irontianta Gkorezi-Ntavela, Dimitra Pappa, Alexandra Bargiota, Antonios Garas, Ioanna Grivea, Alexandros Daponte

**Affiliations:** 1Department of Obstetrics & Gynaecology, University Hospital of Larisa, 41334 Larisa, Greece; gvalasoulis@gmail.com (G.V.); igorezi@hotmail.com (I.G.-N.); garasant@med.uth.gr (A.G.); 2Hellenic National Public Health Organization—ECDC, 15123 Athens, Greece; 3Second Department of Pathology, National and Kapodistrian University of Athens, Attikon University Hospital, 12464 Athens, Greece; apouliak@med.uoa.gr; 4Department of Endocrinology and Metabolic Diseases, Faculty of Medicine, School of Health Sciences, University of Thessaly, 41334 Larissa, Greece; di.pappa@hotmail.com (D.P.); abargio@med.uth.gr (A.B.); 5Department of Pediatrics, University of Thessaly, University Hospital of Larissa, 41334 Larissa, Greece; iogrivea@med.uth.gr

**Keywords:** breastfeeding, individualized intervention, endocrine disorders in pregnancy, midwifery care

## Abstract

Breastfeeding rates remain extremely low in Greece and women with gestational diabetes mellitus and hypothyroidism may experience additional difficulties. The aim of the study was to investigate the effect of a structured individualized lactation educational intervention by a midwife on increasing breastfeeding rates in women with endocrine disorders and low-risk women compared to women receiving standard care, 24 months after delivery. Two-hundred women made up the study population. Half of them were experiencing endocrine pregnancy disorders and 100 women constituted the low-risk pregnancy standard care control group. Women who were breastfeeding exclusively were significantly higher in the midwifery intervention group with endocrine disorders, namely breastfeeding continued at four months (breastfeeding: 20% vs. 12%, exclusive breastfeeding: 50% vs. 26%, *p* = 0.0228), and at six months after childbirth (breastfeeding: 54% vs. 28%, exclusive breastfeeding: 32% vs. 12%, *p* = 0.0011), compared to the standard care control group with endocrine disorder. The low-risk midwifery intervention group breastfed at four months (22% vs. 14%, *p* = 0.0428) and at six months (52% vs. 26%, *p* = 0.0018) at higher rates compared to the standard care control group. In addition, exclusive breastfeeding was significantly higher in the low-risk midwifery intervention group at four months (46% vs. 20%, *p* = 0.0102) and six months (38% vs. 4%, *p* < 0.0001) compared to the standard care control group. This study was the first attempt of a structured midwifery breastfeeding education in Greece and its major contribution reflects a significant positive impact on breastfeeding rates in terms of duration and exclusivity in women with gestational endocrine disorders as well as in low-risk women, and could possibly be applied and instituted in everyday clinical practice to increase the low breastfeeding rates in Greece.

## 1. Introduction

It is well established that breastfeeding has benefits for infants, children, mothers, and public health [1,2,3]. Breastfeeding increases protection against childhood infections, decreases the likelihood for development of type II diabetes mellitus in children and adolescents/young women, as well as reduces the incidence risk of metabolic diseases in children, while exclusive breastfeeding may additionally even reduce the potential risk for future childhood obesity [1,4,5,6,7,8,9].

Therefore, exclusive breastfeeding remains a standard recommendation by the majority of women’s and children’s health organizations, including the World Health Organization (WHO) [10,11]. Evidence shows that despite the fact that many women initiate breastfeeding, few finally meet the recommended goals in terms of duration and exclusivity [12,13,14,15,16].

In spite of the perceived health benefits for both mothers and infants health, significant disparities still exist in the rates of breastfeeding between different countries. In low and middle-income countries, less than 40% (37%) of the infants breastfeed exclusively and for a mean postnatal time period of shorter than six months, whilst in high-income countries this duration appears to be even shorter [1]. Based on the data of the national breastfeeding study conducted in Greece, while 94% of the infants start breastfeeding at the first hour after delivery, they exclusively breastfeed up to the end of the sixth month at a rate of less than 1% [17]. 

### 1.1. Gestational Diabetes Mellitus, Hypothyroidism and Breastfeeding

High maternal weight, pregnancy at late age ranges, and dietary factors such as vitamin B12 deficiency have been linked with increased risk for developing gestational diabetes mellitus (GDM), and that risk is increasing in western living model countries [18,19,20,21]. GDM complications to the fetus and infants include macrosomia, intrauterine growth restriction (IUGR), metabolic syndrome and Type 2 diabetes mellitus. Furthermore, studies have shown that women with GDM breastfed exclusively with even lower rates and for shorter periods of time compared with women not experiencing GDM [22,23,24,25,26,27,28,29]. Overall, a systematic review of the literature reported that GDM was associated with shorter total breastfeeding duration, reduced rates of exclusive breastfeeding, and a higher risk for premature termination of breastfeeding [30].

In addition to GDM, impaired thyroid function in pregnancy may affect the duration of breastfeeding [31]. Detailed research in the particular field has shown that hypothyroidism is associated with repression of breast milk production, although clinical and subclinical hypothyroidism appear to be associated with an increased risk of perinatal complications [32,33,34,35,36]. Women with GDM and hypothyroidism in pregnancy may need additional support in starting and maintaining breastfeeding, and these individuals should be encouraged and supported properly/adequately [4,5]. 

### 1.2. Interventions to Support Breastfeeding

Personalized education and support are likely to increase breastfeeding initiation, exclusivity and duration rates [37,38,39]. Previous studies showed that social factors and targeted intervention in the prenatal or postnatal period might have a positive effect on the onset and duration of breastfeeding, increase confidence in breastfeeding and provide women with adequate knowledge about breastfeeding and increase exclusive breastfeeding [40,41,42,43,44,45,46,47,48,49]. A systematic review and meta-analysis illustrated that all types of support reduced the likelihood of cessation of exclusive breastfeeding and any kind of breastfeeding before four to six weeks (postnatal), as well as the likelihood of cessation of exclusive breastfeeding at six months postnatal, while women who were trained in breastfeeding were 41% more likely to start and continue breastfeeding than women who were not [16,50]. In another meta-analysis, the intervention applied in the prenatal and postnatal period, similar to that in our study, was presented as more effective than the interventions applied, either only in the prenatal period or only in the postnatal period [51].

The aim of the study was to examine the effect of a personalized prenatal and postnatal structured midwifery intervention programme on increasing breastfeeding rates and exclusive breastfeeding, especially in women with endocrine pregnancy disorders and low-risk pregnancies compared to women who received standard hospital care and no structured midwife intervention up to two years after childbirth. 

## 2. Materials and Methods

### 2.1. Study Population

A total population of 200 individuals (100 low-risk pregnant women and 100 pregnant women with GDM and/or hypothyroidism in pregnancy) were included to the final study. The low-risk group women were randomly subdivided into two equal subgroups, (1) the midwifery intervention group, and (2) the standard care control group. The women with GDM and/or hypothyroidism in pregnancy were randomly divided into two equal population subgroups, a midwifery intervention group and a standard care control group. GDM is defined as any degree of glucose intolerance with onset or first recognition during pregnancy and appears to be a temporary condition that occurs in pregnancy, carrying additionally long-term risk of type 2 diabetes development [52]. Hypothyroidism is defined by a lack of thyroid hormones caused by autoimmune thyroid (Hashimoto thyroiditis), iodine deficiency or following surgery or radioiodine therapy [53]. Women substituting the two standard care control subgroups were not obliged to attend personalized breastfeeding classes, except as standard care.

Exclusive breastfeeding was defined according to the WHO recommendations, where an infant receives only breast milk and no other fluids or solids (are given) with the sole exception of vitamins, hydration solutions, minerals or medicines, regardless of whether it is almost exclusive or in combination with a breast milk substitute or other food, liquid or solid [11].

Exclusive breastfeeding and breastfeeding rates were assessed during their stay in the postnatal ward after labour, as well as at the first, fourth and sixth months after childbirth; however, the duration of breastfeeding was estimated additionally at the 12th and 24th month at the follow up visits two years post childbirth. According to the study protocol, all low-risk pregnant women were monitored in the outpatient antenatal clinic and those with GDM and hypothyroidism in the specialty endocrinological antenatal clinic. The study was held at the University Hospital of Larisa, the only tertiary hospital of central Greece in the period from March 2017 to March 2020.

### 2.2. Inclusion & Exclusion Criteria

We included pregnant women speaking the Greek language fluently (thus being consequently able to complete the questionnaires), individuals over 18 years of age, over 24 weeks of gestation at the time of recruitment, and willing to participate in the study, with the intention to be continuously monitored up to the 24th month at the hospital during the antenatal period. Individuals who agreed to participate received detailed information regarding the purpose of the study and signed an informed consent form. We excluded all individuals who did not complete and sign the consent form, and women who refused to complete the socio-demographic data questionnaire. Furthermore, women were allowed to exit of study at any time from recruitment until the end of the study if they wanted to.

### 2.3. Intervention

The midwifery intervention, which included prenatal and postnatal education as well as support for breastfeeding, was carried out individually and took place in two phases. As part of the midwifery intervention programme, in the first phase of the study a preliminary similar self-supplemented questionnaire was given to the midwifery intervention group before the interview and to the standard care control group. Both groups (intervention and standard care control) were focusing on socio-demographic data, the status of the present pregnancy, hereditary diseases, individual history of GDM/hypothyroidism, previous breastfeeding experience, breastfeeding intention, pregnant women’s knowledge, and attitudes and views on breastfeeding. The women in the standard care control and midwifery intervention groups completed the first questionnaire while waiting for their scheduled appointment at the hospital clinic.

After completing the first questionnaire, the women in the low-risk midwifery intervention group as well as women constituting the GDM and gestational thyroid disorder (hypothyroidism) midwifery intervention subgroup were asked to attend an individual breastfeeding counselling session that lasted for three hours. In a specific area of the hospital, both midwifery intervention subgroups underwent individual training for breast-feeding by a specialized midwife. The midwifery intervention took place after the 34th week of gestation and was free of charge. It consisted of prenatal and postnatal breastfeeding training and was based on the WHO’s guidelines on breastfeeding, the 10 steps of successful breastfeeding in infant-friendly hospitals, the international code of marketing of breast milk substitutes, and the scientific guidelines of the Institute of Child Health Greece [54]. The women’s partners were not present during the time of the intervention.

The aim of prenatal education was to inform pregnant women about the importance and the benefits of breastfeeding, demonstrate breastfeeding techniques, develop skills, manage breastfeeding problems, support and empower women to start and consolidate breastfeeding as well as to improve maternal confidence and the self-efficacy on breastfeeding. The 100 women of the two midwifery intervention subgroups (50 pregnant women with GDM and hypothyroidism and 50 low-risk pregnant women) were further informed about the anatomy and physiology of milk production, the importance of exclusive breastfeeding, the breastfeeding on-demand and the importance of the skin-to-skin contact. Participants were asked to share their concerns with the midwife and their personal perceptions of breastfeeding were discussed in detail (Table 1).

In terms of increased awareness, we used leaflets of the Institute of Child Health-Greece, medical animation dolls, and presentations. An illustrated breastfeeding information pamphlet (of the Institute of Child Health—Alcyone) was given to each pregnant woman after the end of the intervention, as well as the telephone number of the midwife in charge to answer any possible questions and problems related to breastfeeding. The second phase of the midwifery intervention consisted of a planned visit on the second to third day after childbirth during their stay in the postnatal ward, focusing on reviewing the process of breastfeeding, the attachment of the newborn, the enhancement of the mother’s self-efficacy, the recognition of signs of nutritional adequacy of the new-born, the resolution of breastfeeding problems and direct support during breastfeeding observations. We observed the position and attitude of the mother while holding the baby, the method of supporting the breast, the infant’s latching on to breast when suckling, and signs of effective suckling and gave specific instructions and advice (see Table 1 and for more details Appendix A).

The subsequent phases of the survey were carried out by telephone communication from the midwife in charge who carried out the intervention at specified intervals: three weeks, four months, six months, and first and second year after birth, in order to record the frequency and indicators of breastfeeding as well as to empower women, support them for any difficulties of breast feeding and provide information.

### 2.4. Data Collection

Data collection included a prenatal interview, a postnatal interview and telephone interviews. The prenatal interview covered demographic data information, previous breastfeeding experience, breastfeeding intention, and information about the current pregnancy. The postnatal interview referred to data on childbirth, the initial time of the first breastfeeding, the type of breastfeeding, the condition of the infant, the administration of modified milk during their hospitalization and health care professional’s support. It was completed on the second to third day postpartum during hospitalization by the intervention midwife in charge. The telephone interviews were conducted at four weeks, 12 weeks, six months, 12 and 24 months respectively for the assessment of feeding infants. The number of scheduled telephone communications was the same for all the women of the intervention groups, with the duration of telephone communication likely to be modified on the basis of women’s queries. During the course of the survey it was possible for participants to contact the intervention midwife in charge for any questions beyond the scheduled telephone communication.

#### 2.4.1. Socio-Demographic Data Questionnaire

The socio-demographic data questionnaire which was designed by the principal investigator of the study, and approved by the ethics committee of the Institution including general personal information, mother’s medical history, family’s history, presence of breastfeeding experience and intention to breastfeed.

#### 2.4.2. Characteristics under Examination

Maternal self-reporting was used for breastfeeding evaluation and in every follow-up interview and women were asked to report the status of breastfeeding. The dependent variables used to assess the effectiveness of intervention based on WHO definitions were: (1) the initiation of breastfeeding (2) the duration of any breastfeeding and (3) the duration of exclusive breastfeeding. The duration of breastfeeding was estimated at 1 month, 4 months, 6 months, 12 and 24 months after childbirth. Selected variables were examined for relations with the duration of breastfeeding at the 1st, 4th and 6th month after childbirth. The variables included professional support, family support and parameters such as the counselling by telephone from midwife and existence of gestational pathology (GDM and/or hypothyroidism).

#### 2.4.3. Ethical Issues

All women were informed about the scope of the study and were asked to undersign a consent form before entering the study. The study’s protocol has been approved by the Scientific and Ethics Committee of the University Hospital of Larissa (Code: 29488/11-7-2016, Date: 11 July 2016). The questionnaires were accompanied by an informative note for the purpose of the study and an undersigned statement by the individuals that they had the right, at any time, to discontinue their participation in the survey. 

#### 2.4.4. Statistical Analysis

Data were collected in Microsoft Excel spreadsheets and subsequent statistical analysis was performed via the SAS for Windows 9.4 software platform (SAS Institute Inc., Cary, NC, USA). For the arithmetic data (such as age) descriptive values were expressed as mean ± standard deviation (SD) while for the categorical data (for example education level) as frequency and the relevant percentages. Comparisons between groups were performed using the chi-square test, for comparisons between two groups the Odds Ratios (OR) and the relevant 95% Confidence Intervals (CI) were calculated. The significance level (*p*-value) for the complete study was set to *p* < 0.05.

## 3. Results

### 3.1. Description Sample

The Socio-demographic and pregnancy related characteristics of the sample as well as the breastfeeding of other children are presented in Table 2.

### 3.2. Indicators and Duration of Breastfeeding

#### Midwifery Structured Intervention in Low & High-Risk Populations

First, the analysis of the duration of exclusive or any breastfeeding in the high-risk group of women (women with GDM or hypothyroidism) was carried out, comparing the rates of exclusive breastfeeding or any breastfeeding between both subgroups (intervention group vs. control group). The results underlined: (1) improved breastfeeding rates in the intervention group with gestational endocrine disorder to breastfeed for four to six months, specifically 10 women (20% in the four months group) and six women (representing 12% within the four months group), breastfed for four to six months (OR: 3.8, 95% CI: 1.2–12.8, *p* = 0.0358, as compared to no breastfeeding and breastfeeding for three weeks, Table 3); (2) At six months after childbirth the 54% individuals of the intervention group with endocrine pregnancy disorder (54 vs. 28%) continued to breastfeed (OR: 4.5, 95% CI: 1.8–11.1 as compared to the group of women that did not breastfed of breastfed for three weeks combined, *p* = 0.0020, Table 3); (3) Women in the intervention group with gestational endocrine disorder breastfed exclusively at higher rates (50% vs. 26%, OR: 2.8, 95% CI: 1.2–6.6) compared to standard care controls (*p* = 0.0228) at four months (Table 3), with the differences between the intervention group with endocrine disorder to appear remaining statistically significant at six months after delivery (32% vs. 12%, *p* = 0.0011) compared to controls (OR: 3.5, 95% CI: 1.2–9.8, *p* = 0.0283, Table 3); (4) More women, allocated to the intervention group, were continuing to breastfeed for 12 months after delivery compared to controls (32%, vs. 18%, OR: 2.1, 95% CI: 0.8–5.5, *p* = 0.1060, Table 3); and (5) the particular breastfeeding rate was found to be doubled after two years of childbearing (18% vs. 8%, OR: 2.5, 95% CI: 0.7–8.8, *p* = 0.2336 Table 3).

Similarly, the analysis of the duration of exclusive or any breastfeeding in the group of women with low-risk pregnancies was performed where the rates of exclusive breastfeeding or any breastfeeding were compared between both subgroups (intervention group vs. control group). According to the results: (a) at the 4 months postnatal evaluation, more women allocated to the low-risk intervention group were breastfeeding compared to those in the low-risk control group (22% vs. 14%, OR: 3.6, 95% CI: 1.1–11.4, *p* = 0.0428, Table 3); (b) at the six month postnatal evaluation, women in the low-risk intervention group were more likely to breastfeed (52% vs. 26%, OR: 4.6, 95% CI: 1.8–11.7, *p* = 0.0018, Table 3); (c) at the four month postnatal visit, the exclusivity of breastfeeding was significantly higher (46% vs. 20%, OR: 3.4, 95% CI: 1.4–8.3, *p* = 0.0102, Table 3); (d) the rates continued to increase significantly at six months at the intervention group (38% vs. 4%, OR: 14.7, 95% CI: 3.2–67.6, *p* < 0.0001, Table 3); (e) more women in the intervention group were breastfeeding for 12 months (30% vs. 10%, OR: 3.9, 95% CI: 1.3- 11.6, *p* = 0.0228, Table 3); and (f) The rate was estimated to be more than doubled at two years (6% vs. 2%, OR: 3.1, 95% CI: 0.3–31.1, *p* > 0.05, Table 3).

### 3.3. Relation of Variables for Duration of Breastfeeding

Women who received postpartum counselling from a midwife breastfed for 6 months at higher rates (OR: 6.3, 95% CI: 3.3–12.0, Table 4) compared to the women that breastfed for less time (chi-square = 34.6, *p* < 0.0001). Rates of breastfeeding in women who received telephone support/counselling in postpartum period from a midwife, increased statically significantly at six months (OR: 5.1, 95% CI: 2.7–9.7) compared to those that did not receive telephone support (chi-square: 26.8, *p* < 0.0001, Table 4). Family support seemed to affect also the duration of breastfeeding, specifically, women with family support had higher odds (OR: 5.7, 95%CI: 1.3–25.8) as compared to those women without family support (chi-square: 6.3, *p* = 0.0122, Table 4). It appeared that women who breastfeed exclusively from the first days continued to breastfeed exclusively at six months (OR: 12.7, 95% CI: 3.6–44.3) as compared to those that used additional formula (chi-square: 23.22, *p* < 0.0001, Table 4). The occurrence of GDM, or hypothyroidism in the present pregnancy, did not appear to affect the duration of breastfeeding (*p* > 0.05, Table 4).

## 4. Discussion

The aim of our study was to examine the possible effect of a structured individualized prenatal and postnatal lactation educational intervention by a specialty midwife, on improving breastfeeding rates and indicators in women with gestational endocrine disorders as well as in a low-risk hospital population, compared to women who received standard care in a time manner of 24 months after delivery. The results of that effort showed that individualized structured prenatal and postnatal breastfeeding education by a specialty midwife might lead to increased improved rates of duration of any type of breastfeeding and exclusive breastfeeding in women with gestational endocrine disorders and in low-risk Greek women as well as should therefore be included as standard practice.

Midwifes could influence and increase breastfeeding rates by training women how to understand the whole process and nature of breastfeeding, outcomes that have been proved similarly and in our study using our intervention protocol [15,55,56,57,58]. More specifically, individualized education in low-risk women and women with endocrine diseases of midwifery staff as in our study, proved to be effective in accordance with the published evidence, as it reduces the likelihood of cessation of breastfeeding, and increases women’s knowledge, skills and confidence in breastfeeding [39,59].

The results of the particular study stay in line with already established data in other countries demonstrating the effectiveness of midwife breastfeeding interventions in low-risk women. Published evidence has already linked several similar antenatal interventions’ effectiveness in increasing breastfeeding rates postnatally [60,61,62,63]. Several studies so far have shown that prenatal and postnatal counselling of low-risk women and their families increases the rates of exclusive breastfeeding as well as the total duration of breastfeeding and appears to have a positive effect on empowering women to breastfeed [44,64,65]. Similarly, studies concluding that interventions combining personalized education to prenatal and postnatal support (postpartum clinic visits) as well as telephone communication as in our study are most likely to increase any type of breastfeeding and exclusive breastfeeding duration and rates [37,66,67,68]. Based on the study of Schreck et al., prenatal intervention in low-risk women increased breastfeeding initiation rates, while women who received both prenatal and postnatal support were found to breastfeed at higher rates at six months after childbearing [69]. A systematic review has shown that intervention in low-risk women could increase initiation rates as well as the exclusive breastfeeding rates to six months postpartum [37]. It must be emphasized that studies demonstrated that individualized training, as in our protocol, seemed to be more effective compared to group intervention or telephone support exclusively [70,71]. A meta-analysis by Kim et al. demonstrated that intervention in terms of exclusive breastfeeding effectiveness, for the period of six months after delivery, appeared to improve breastfeeding rates significantly and presented as most effective the prenatal and postnatal intervention [51]. Based on our results and the above-mentioned literature, our group also believes that individualized prenatal and postnatal breastfeeding counselling by a specialty midwife could increase the duration of breastfeeding and exclusive breastfeeding in low-risk women and should be introduced in everyday clinical practice.

### Breastfeeding in Endocrine Disorders in Pregnancy

The results of our study appear to be consistent with findings from other studies where authors demonstrated the effectiveness of breastfeeding interventions in women with endocrine disorders of pregnancy. A study by You et al. revealed that individualized intervention has a positive effect on breastfeeding rates as well as on exclusive breastfeeding in women with GDM [72]. In line with our results, a study by Tawfik et al. illustrated that more women with GDM after receiving intervention compared with similar standard care controls reported exclusive breastfeeding for six weeks postpartum, concluding that individualized prenatal and postnatal counselling in women with GDM is definitely effective in increasing the duration and exclusivity of breastfeeding [73,74,75]. Similarly, and in concordance to our findings, a study by Stuebe et al. concluded that personalized education in women with endocrine disorder from a specialty midwife in charge could be effective, as it reduces the likelihood of cessation of breastfeeding or the introduction of formula, makes women more likely to breastfeed throughout the study period, and makes them more likely to breastfeed exclusively [58]. 

The major limitation of this study appears to be that women in the intervention and control groups participated voluntarily and therefore might have had a predisposition to breastfeed, thus invoking the risk of bias. In addition, the participants of the study were recruited from only one healthcare unit/hospital in a large geographical region of Thessaly, and particularly in the city of Larissa, limiting in a way the safe establishment of clear outcomes in the general population. The study did not include home visits during the postpartum period, but only counselling/training in the clinic and telephone breastfeeding support after discharge. Future studies would benefit by including additional variables, such as the husband’s training.

## 5. Conclusions

In conclusion, individualized structured prenatal and postnatal breastfeeding education by a specialty midwife improved breastfeeding frequency in women with endocrine disorders in pregnancy as well as in low-risk women. The major contribution of the study reflects a significant positive impact on breastfeeding rates of individualized structured prenatal and postnatal breastfeeding education by a specialty midwife in terms of duration and exclusivity (of breastfeeding) in women with gestational endocrine disorders as well as in the low-risk female population. 

This was the first study investigating breastfeeding rates in central Greece with respect to a population and its specific cultural features, taking place in a tertiary hospital covering an area inhabited by 1,000,000 citizens. The beneficial effects of this structured midwifery intervention could project evolving regional hospital guidelines including such a structured custom-made midwifery intervention in the everyday midwifery clinical practice of standard care (based on our results with local population). The unfortunately overall low breastfeeding rates in Greece increased after the intervention and reached the target of exclusive breastfeeding for an infant up to six months of age, with continued breastfeeding along with appropriate complementary foods up to two years of age or even beyond. Future similar studies in the different regions could improve the breastfeeding rates in the Greek population with the help of appropriately trained midwives.

## Figures and Tables

**Table 1 ijerph-18-11359-t001:** The phases and the issues of the structured midwifery intervention.

Midwifery Intervention	Type
1. Personalized prenatal midwifery program (after 34 weeks of gestation)	Structured midwifery education-support
2. Personalized postnatal structured midwifery program (one visit postpartum at 2nd–3rd day)	Structured midwifery education and support
3. Telephone communication from the midwife (at 3 weeks, 4 months, 6 months, first and second year after birth)	Midwifery Telephone support and counselling

**Table 2 ijerph-18-11359-t002:** Baseline descriptive characteristics of the study population.

Characteristics	Values
Age (mean ± SD)	32.4 ± 13.2
Marital status	
Married	177 (88.5%)
Divorced	3 (1.5%)
Single	20 (10.0%
Living place	
Village/small town	81 (40.5%)
City with population <150,000	52 (26.0%)
City with population >150,000	67(33.5%)
Education level	
High school	89 (44.7%)
Higher education	73 (36.7%)
Master of Science	12 (6.0%)
Employment	
Private employees	53 (26.5%)
Civil servants	17 (8.5%)
Self-employed	36 (18%)
Domestically employed	33 (16.5%)
Unemployed	58 (29%)
Nationality	
Greek	174 (87%)
Other	26 (13%) [Albanian: 13 (6.5%)]
Has other children	88 (44%)
Number of other children	
1	74 (37%)
2	0 (0%)
≥3	14 (7%)
Religion	
Christian orthodox	184 (92%)
Muslim	12 (6%)
Other	4 (2%)
Breastfeeding of other children	69 (34.5%)

**Table 3 ijerph-18-11359-t003:** Number of cases and percentage of breastfeeding duration in the control group and experimental group in each evaluation period. Each cell shows: the number of cases and the percentage within the group (row percentage). BF: Breastfeeding.

	Duration				BF at 4 Months			BF at 6 Months				BF for 1 Year	BF for 2 Years
Group/Midwifery Intervention	No BF	3 Weeks	4 Months	6 Months	No BF	Exclusive	Any Type of BF	No BF	Exclusive	Any Type of BF	Complementary Feeding		
High Risk/Intervention	0 (0%)	13 (26%)	10 (20%)	27 (54%)	0 (0%)	25 (50%)	25 (50%)	23 (46%)	16 (32%)	5 (10%)	6 (12%)	16 (32%)	9 (18%)
High Risk/No-Intervention	4 (8%)	26 (52%)	6 (12%)	14 (28%)	4 (8%)	13 (26%)	33 (66%)	34 (68%)	6 (12%)	2 (4%)	8 (16%)	9 (18%)	4 (8%)
Low Risk/Intervention	1 (2%)	12 (24%)	11 (22%)	26 (52%)	1 (2%)	23 (46%)	26 (52%)	24 (48%)	19 (38%)	2 (4%)	5 (10%)	15 (30%)	3 (6%)
Low Risk/No-Intervention	5 (10%)	25 (50%)	7 (14%)	13 (26%)	5 (10%)	10 (20%)	35 (70%)	37 (74%)	2 (4%)	6 (12%)	5 (10%)	5 (10%)	1 (2%)
Total	10 (5%)	76 (38%)	34 (17%)	80 (40%)	10 (5%)	71 (35.5%)	119 (59.5%)	118 (59%)	43 (21.5%)	15 (7.5%)	24 (12%)	45 (22.5%)	17 (8.5%)

**Table 4 ijerph-18-11359-t004:** Factors associated with breastfeeding continuation.

Variable	First Days (*n* = 76)	≤4 Months (*n* = 34)	≤6 Months (*n* = 80)	*p*
Support for breastfeeding after returning home by a midwife	13 (17.1%)	18 (52.9%)	57 (71.3%)	<0.0001
Support for breastfeeding after returning at home by telephone from midwife	11 (14.5%)	12 (35.3%),	46 (57.5%),	<0.0001
Family support for breastfeeding after returning at home	63 (82.9%)	33 (97.1%),	78 (97.5%),	0.002
Exclusive breastfeeding (vs. breastfeeding and formula)	2 (2.6%)	1 (2.9%)	21 (26.3%)	<0.0001
Diabetes during pregnancy	21 (27.6%)	12 (35.3%)	26 (32.5%)	0.6773
Thyroid disorder during pregnancy	23 (30.3%)	7 (20.6%)	21 (26.3%)	0.5643

## Data Availability

Data are available from the corresponding author upon a reasonable request.

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
