# Peer review of "The Effect of a Structured Individualized Educational Intervention on Breastfeeding Rates in Greek Women"

_ijerph, 2021, doi:10.3390/ijerph182111359_

Round 1

Reviewer 1 Report

Dear authors,

Thank you for the effort made to try to improve your study. It is appreciated the work done and a clear improvement over the previous version. However, in order to have more options for publication, it would be convenient to review some aspects a second time.

The introduction presents clear improvements in its structure and in the references of the studies consulted. For example, an notable improvement has been to specify thyroid dysfunction, making reference to hypothyroidism. Nevertheless, it would be beneficial to review some issues:

In the third paragraph of the introduction, why has the term newborn been replaced by neonate? It could be correct according to the context of the sentence but it is important to keep in mind that neonate refers to a baby of 28 weeks or less who has been born preterm, at term or born after the ninth month of pregnancy. Check if the terms fit according to the bibliography consulted.

In the first subsection of the introduction, the authors name Gestational Diabetes Mellitus, it would be convenient to review when it is named for the first time and include the acronym (GDM) from that moment on.

Regarding the methodology, the distribution of the groups being compared (control group and experimental group) and the randomization of the sample are clear. However, I would like to make some suggestions.

The WHO reference to the definition of Breastfeeding is missing.

In my opinion it would be clearer to explain the condition of GDM and hypothyroidism as definition of these pathologies instead of criteria since, it is confusing. However, if they prefer to call these terms as criteria they should include them in the information on inclusion and exclusion criteria in section 2.2 and expose it in a table for simplification.

It is not necessary for the authors to specify the N and the type of population that makes up each subgroup of the control and experimental groups, as this is repetitive.

In line 204 the word breast-feeding appears twice, it would be convenient to unify the term.

In this second revision, the authors have not indicated the duration of the study, when data collection started and ended. This is an important error that should be corrected.

With regard to the results, it would be advisable to revise some aspects.

For example, Table 2 should include a footnote specifying the meaning of "M.Sc.".

The title of table 3 is confusing. In my opinion, it would be more recommendable, for example, "number of cases and percentage of breastfeeding duration in the control group and experimental group in each  evaluation period".

In addition to the above, the most important thing is to revise the wording of the results because, although it has improved with respect to the previous version of the manuscript, it is still somewhat confusing due to the multiple comparisons. I suggest the authors revise 3.21 and use simpler structures. For example:

“First, the analysis of the duration of exclusive or any breastfeeding in the high-risk group of women (women with GDM or hypothyroidism) was carried out, comparing the rates of exclusive breastfeeding or any breastfeeding between both subgroups (intervention group vs. control group) ..... (describe these results).

Similarly, the analysis of the duration of exclusive or any breastfeeding in the group of women with low-risk pregnancies was performed where the rates of exclusive breastfeeding or any breastfeeding were compared between both subgroups (intervention group vs control group).... (describe these results)”.

I hope you agree with the proposed suggestions and that they will help you to improve your work.

Yours sincerely,

Author Response

Reviewer 1

Comment 1: Thank you for the effort made to try to improve your study. It is appreciated the work done and a clear improvement over the previous version. However, in order to have more options for publication, it would be convenient to review some aspects a second time.

The introduction presents clear improvements in its structure and in the references of the studies consulted. For example, a notable improvement has been to specify thyroid dysfunction, making reference to hypothyroidism. Nevertheless, it would be beneficial to review some issues:

Authors’ actions: Thank you for reviewing again our manuscript. We strongly believe that the updated version, after your second revision and recommendations, has been improved adequately.

Comment 2: In the third paragraph of the introduction, why has the term newborn been replaced by neonate? It could be correct according to the context of the sentence but it is important to keep in mind that neonate refers to a baby of 28 weeks or less who has been born preterm, at term or born after the ninth month of pregnancy. Check if the terms fit according to the bibliography consulted.

Authors’ actions: We have changed back this term to newborn as requested. However, according to the WHO definition “A new-born infant, or neonate, is a child under 28 days of age” (https://www.who.int/westernpacific/health-topics/newborn-health)

Comment 3: In the first subsection of the introduction, the authors name Gestational Diabetes Mellitus, it would be convenient to review when it is named for the first time and include the acronym (GDM) from that moment on.

Authors’ actions: We agree with your proposal; we have replaced everywhere “Gestational Diabetes Mellitus” by GDM. Thank you.

Comment 4: Regarding the methodology, the distribution of the groups being compared (control group and experimental group) and the randomization of the sample are clear. However, I would like to make some suggestions.

The WHO reference to the definition of Breastfeeding is missing.

Authors’actions: We have added the relative reference. Thank you. https://apps.who.int/nutrition/topics/exclusive_breastfeeding/en/index.html

Comment 5: In my opinion it would be clearer to explain the condition of GDM and hypothyroidism as definition of these pathologies instead of criteria since, it is confusing. However, if they prefer to call these terms as criteria they should include them in the information on inclusion and exclusion criteria in section 2.2 and expose it in a table for simplification.

Authors’ actions: Thank you for your remark. GDM and hypothyroidism have been defined in the revised version manuscript.

Comment 6: It is not necessary for the authors to specify the N and the type of population that makes up each subgroup of the control and experimental groups, as this is repetitive.

Authors’ actions: We agree with you comment. We have removed the repetitive presentation of the number of cases in each study subgroup in the materials and methods section. Thank you.

Comment 7: In line 204 the word breast-feeding appears twice, it would be convenient to unify the term.

Authors’ actions: Thank you for pointing, it was removed.

Comment 8: In this second revision, the authors have not indicated the duration of the study, when data collection started and ended. This is an important error that should be corrected.

Authors’ actions: The study duration as well as start and end points have been added in the revised version.

Comment 9: With regard to the results, it would be advisable to revise some aspects.

For example, Table 2 should include a footnote specifying the meaning of "M.Sc.".

The title of table 3 is confusing. In my opinion, it would be more recommendable, for example, "number of cases and percentage of breastfeeding duration in the control group and experimental group in each evaluation period".

Authors’ actions: M.Sc. was replaced by “Master of Science”, since this was the sole time appearing in the manuscript. The title was replaced as you propose, thank you!

Comment 10: In addition to the above, the most important thing is to revise the wording of the results because, although it has improved with respect to the previous version of the manuscript, it is still somewhat confusing due to the multiple comparisons. I suggest the authors revise 3.2 and use simpler structures. For example:

“First, the analysis of the duration of exclusive or any breastfeeding in the high-risk group of women (women with GDM or hypothyroidism) was carried out, comparing the rates of exclusive breastfeeding or any breastfeeding between both subgroups (intervention group vs. control group) ..... (describe these results).

Similarly, the analysis of the duration of exclusive or any breastfeeding in the group of women with low-risk pregnancies was performed where the rates of exclusive breastfeeding or any breastfeeding were compared between both subgroups (intervention group vs control group).... (describe these results)”.

Authors’ actions: Section 3.2 was revised as proposed, moreover we numbered the individual results as a), b), c) etc and wording was simplified.

Comment 11: I hope you agree with the proposed suggestions and that they will help you to improve your work.

Authors’ actions: Thank you for your time and suggestions, they were really helpful and improved manuscript’s quality.

Reviewer 2 Report

Dear Authors.

It was my pleasure to read your article.

I have some questions:

  1. I do not fully understand some parts of table 2:
    1.  what do you mean using the phrase "Diligence of other children"?
    2. what do you exactly mean: Number of other children - One 74 (37%); ≥3 14 (7%)?

Well performed midwife' consultation is crucial for breastfeeding mothers.

This article is so interesting and value because usually patients with diabetes mellitus and thyroid disfunction breastfeed their newborn babies shorther and they have more difficulties than healthy indyviduals.

This article prove that the midwives as professionals have the crucial impact for mothers. Breastfeeding is undisputable the basis of the well-being for offspring and society.

Author Response

Reviewer 2

Comment 1: It was my pleasure to read your article.

I have some questions:

  1. I do not fully understand some parts of table 2:
    1.  what do you mean using the phrase "Diligence of other children"?
    2. what do you exactly mean: Number of other children - One 74 (37%); ≥3 14 (7%)?

Authors’ actions: We have amended the table content in order to resolve such issues, thank you for pointing.

Comment 2: Well performed midwife' consultation is crucial for breastfeeding mothers.

This article is so interesting and value because usually patients with diabetes mellitus and thyroid disfunction breastfeed their newborn babies shorther and they have more difficulties than healthy individuals.

This article prove that the midwives as professionals have the crucial impact for mothers. Breastfeeding is undisputable the basis of the well-being for offspring and society.

Authors’ actions: No action, thank you!

Reviewer 3 Report

Dear authors,
I think your discussion in particular need some modifications in relation to the  similar literature. I think the intervention of midwife need more clarification

Author Response

Reviewer 3

Comment 1: I think your discussion in particular need some modifications in relation to the similar literature. I think the intervention of midwife need more clarification

Authors’ actions: Thank you for your remark. A summary of the phases and the issues of the structured midwifery intervention of our study are presented in table 1 in the manuscript. Midwifery Intervention In detail is also described in the following table which was also added in Appendix 1.

MIDWIFERY INTERVENTION

TYPE

ISSUES

1.     Personalized prenatal structured midwifery education and support to low risk pregnant and women with endocrine disorder (one session for 2 hours after 34 w gestation)

Structured midwifery education-support

a.     the benefits of breast-feeding, the anatomy and physiology of milk production

b.     the breastfeeding positions, the proper breastfeeding latch, the importance of breastfeeding in the 1st hour after birth the breastfeeding on-demand, the importance of the skin-to-skin contact and the value of rooming in instructions for women diet, management and treatment the difficulties of breastfeeding to improve maternal confidence and the self-efficacy on breastfeeding

2.     Personalized postnatal structured midwifery education and support to low risk pregnant and women with endocrine disorder (one visit on the 2nd-3rd day after childbirth during their stay in the postnatal ward)

Structured midwifery education and support

a.     the position and attitude of mother holding baby, method of supporting breast infant’s latching on to breast when suckling, sings of effective suckling, the resolution of breastfeeding problems and direct support the recognition of signs of nutritional adequacy of the new-born

3.     Telephone communication from the midwife in charge who carried out the intervention at specified intervals (3 weeks, 4 months, 6 months, first and second year after birth)

Midwifery Telephone support

a.     To empower women, support them for any difficulties of breast-feeding and provide information

Furthermore, the following studies have also added as references in our study:

  1. Dagla M, Mrvoljak-Theodoropoulou I, Vogiatzoglou M, Giamalidou A, Tsolaridou E, Mavrou M, Dagla C, Antoniou E. Association between Breastfeeding Duration and Long-Term Midwifery-Led Support and Psychosocial Support: Outcomes from a Greek Non-Randomized Controlled Perinatal Health Intervention. Int J Environ Res Public Health. 2021 Feb 18;18(4):1988. doi: 10.3390/ijerph18041988. PMID: 33670797; PMCID: PMC7922856.
  2. Rm MS, Rn EW, Rn JL, Rm AB. The supporting role of the midwife during the first 14 days of breastfeeding: A descriptive qualitative study in maternity wards and primary healthcare. Midwifery. 2019 Nov;78:50-57. doi: 10.1016/j.midw.2019.07.016. Epub 2019 Jul 15. PMID: 31357116
  3. Fu IC, Fong DY, Heys M, Lee IL, Sham A, Tarrant M. Professional breastfeeding support for first-time mothers: a multicentre cluster randomised controlled trial. BJOG. 2014 Dec;121(13):1673-83. doi: 10.1111/1471-0528.12884. Epub 2014 May 26. PMID: 24861802.

This manuscript is a resubmission of an earlier submission. The following is a list of the peer review reports and author responses from that submission.

Round 1

Reviewer 1 Report

The presented study tackles an issue of The Effect of a Structured Individualised Prenatal and Postnatal Educational Intervention by a Midvife on Breastfeeding Rates in Low Risk Females and Women  With Endocrine Disorders. The study was conducted reliably with appropriate selection of tests. Overall, I think that this article should be published, however after reedition the text according to Instructions for Authors and major revision.

The major issue:

You can’t include in one subjets all patients with diabetes and thyroid diseases (especially hyperthyroidism with hypothyroidism and levels of TSH over 2,5). They have different aetiology and different influence on breasfeeding. I suggest limiting the study to hypothyroidism and GDM and dividing them in separate groups.

Some other  issues require complementary information:

  1. The title -I suggest capitalising the title
  2. Abstract- Is too long. I suggest shortening it according to guidelines for authors
  3. Keywords- It’s too many keywords. I suggest changing it according to guidelines for authors
  4. Verse 73-76- I suggest providing the bibliography for that statements.
  5. Verse 87-89- I suggest including webpade in the bibliogrphy not in the text
  6. Verse 99- I suggest including a few information about neonatal complications of GDM
  7. Verse 171- I suggest icluding the information about gestational age of delivery ( preterm neonates have a big problem with sucking)
  8. Verse 207 - - I suggest including webpade in the bibliogrphy not in the text
  9. Table 1- is unclear. I suggest not doubling the information. Write it in the text or simply in the table.
  10. Tables 4,5,6,7 are unclear- I suggest simplifying.
  11. I suggest deleting the rows „Total” from the tables.
  12. I suggest including in the Discussion the information why breastfeeding rates aren’t high.
  13. I suggest including information concerning the strengths and limitations of the study.
  14. Many bibliographies are a little I suggest updating the bibliography ( 2019 and 2020,2021) e.g.
  • Effects of breastfeeding education based on the self-efficacy theory on women with gestational diabetes mellitus: A CONSORT-compliant randomized controlled trial.

You H, Lei A, Xiang J, Wang Y, Luo B, Hu J.

  • Gestational Diabetes and Breastfeeding Outcomes: A Systematic Review.

Nguyen PTH, Pham NM, Chu KT, Van Duong D, Van Do D.

  • Gestational Diabetes Mellitus Reduces Breastfeeding Duration: A Prospective Cohort Study.

Nguyen PTH, Binns CW, Nguyen CL, Ha AVV, Chu TK, Duong DV, Do DV, Lee AH.

  • A systematic review of infant feeding food allergy prevention guidelines - can we AGREE?Vale SL, Lobb M, Netting MJ, Murray K, Clifford R, Campbell DE, Salter SM.
  • Risk factors for the lack of adherence to breastfeeding.Turke KC, Santos LRD, Matsumura LS, Sarni ROS.
  • Breastfeeding promotion and support: a quality improvement study.Menichini D, Zambri F, Govoni L, Ricchi A, Infante R, Palmieri E, Galli MC, Molinazzi MT, Messina MP, Putignano A, Banchelli F, Colaceci S, Neri I, Giusti A.
  • Maternity ward staff perceptions of exclusive breastfeeding in Finnish maternity hospitals: A cross-sectional study.Hakala M, Kaakinen P, Kääriäinen M, Bloigu R, Hannula L, Elo S.
  • Too little and too late. Initiation of breast feeding in Odisha, India: An observational study.Kuchi S, Sahu S, John J.
  • Increasing the exclusive breastfeeding rate in a private hospital in UAE through quality improvement initiatives. Kaushal M, Sasidharan K, Kaushal A, Augustine P, Alex M.

Author Response

Reviewer 1

Comment 1:  The presented study tackles an issue of The Effect of a Structured Individualised Prenatal and Postnatal Educational Intervention by a Midwife on Breastfeeding Rates in Low Risk Females and Women with Endocrine Disorders. The study was conducted reliably with appropriate selection of tests. Overall, I think that this article should be published, however after reedition the text according to Instructions for Authors and major revision.

The major issue:

You can’t include in one subjets all patients with diabetes and thyroid diseases (especially hyperthyroidism with hypothyroidism and levels of TSH over 2,5). They have different aetiology and different influence on breastfeeding. I suggest limiting the study to hypothyroidism and GDM and dividing them in separate groups.

Authors’ actions: Thank you for your comment. All the individuals that were analysed in the submitted version had hypothyroidism. There were few hyperthyroidism patients that were not included in the final analysis. By mistake it was not mentioned at the description of the study population.

Comment 2: The title -I suggest capitalising the title.

Authors’ actions: We have written the title of the manuscript according to the journal guidance. Thank you.

Comment 3: Abstract- Is too long. I suggest shortening it according to guidelines for authors

Authors’ actions: We have re-written the abstract of the manuscript according to the journal guidelines. Abstract was also modified due to some changes in the analysis, however, without affecting the results. Thank you.

Comment 4: Keywords- It’s too many keywords. I suggest changing it according to guidelines for authors

Authors’ actions: We have updated the keywords. Thank you.

Comment 5: Verse 73-76- I suggest providing the bibliography for that statements.

Authors’ actions: Thank you for your comment. We have added the following bibliography to the references section.

Sriraman NK, Kellams A.J Breastfeeding: What are the Barriers? Why Women Struggle to Achieve Their Goals. Womens Health (Larchmt). 2016 Jul;25(7):714-22. doi: 10.1089/jwh.2014.5059. Epub 2016 Apr 25.PMID: 27111125 Review.

Kellams AL, Gurka KK, Hornsby PP, Drake E, Conaway MR. A Randomized Trial of Prenatal Video Education to Improve Breastfeeding Among Low-Income Women. Breastfeed Med. 2018 Dec;13(10):666-673. doi: 10.1089/bfm.2018.0115. Epub 2018 Oct 23.

Yılmaz E, Doğa Öcal F, Vural Yılmaz Z, Ceyhan M, Kara OF, Küçüközkan T. Turk Early initiation and exclusive breastfeeding: Factors influencing the attitudes of mothers who gave birth in a baby-friendly hospital. J Obstet Gynecol. 2017 Mar;14(1):1-9. doi: 10.4274/tjod.90018. Epub 2017 Mar 15.

Vehling L, Chan D, McGavock J, Becker AB, Subbarao P, Moraes TJ, Mandhane PJ, Turvey SE, Lefebvre DL, Sears MR, Azad MB. Exclusive breastfeeding in hospital predicts longer breastfeeding duration in Canada: Implications for health equity. Birth. 2018 Dec;45(4):440-449. doi: 10.1111/birt.12

Balogun OO, O'Sullivan EJ, McFadden A, Ota E, Gavine A, Garner CD, Renfrew MJ, MacGillivray S. Cochrane Database Interventions for promoting the initiation of breastfeeding. Syst Rev. 2016 Nov 9;11(11):CD001688. doi: 10.1002/14651858.CD001688.pub3.

Cohen SS, Alexander DD, Krebs NF, Young BE, Cabana MD, Erdmann P, Hays NP, Bezold CP, Levin-Sparenberg E, Turini M, Saavedra JM.J Factors Associated with Breastfeeding Initiation and Continuation: A Meta-Analysis. Pediatr. 2018 Dec;203:190-196.e21. doi: 10.1016/j.jpeds.2018.08.008. Epub 2018 Oct 4.

Comment 6: Verse 87-89- I suggest including webpage in the bibliography not in the text.

Authors’ actions: Thank you for your comment. We have put webpage to the references.

Comment 7: Verse 99- I suggest including a few information about neonatal complications of GDM

Authors’ actions: Thank you for your remark. We have included the requested information within the text.

Comment 8: Verse 171- I suggest including the information about gestational age of delivery (preterm neonates have a big problem with sucking)

Authors’ actions: Thank you for your remark. We did not have any preterm neonate.

Comment 9: Verse 207 - - I suggest including webpade in the bibliogrphy not in the text

Authors’ actions: Thank you for your comment. We have put webpage to the references.

Comment 10: Table 1- is unclear. I suggest not doubling the information. Write it in the text or simply in the table.

Authors’ actions: We have updated and modified Table 1 in Table 1.1 and Table 1.2

Comment 11: Tables 4,5,6,7 are unclear- I suggest simplifying.

Authors’ actions: Thank you for your remark. Tables 3 to 7 are now unified in a single table with less information (for example only row percentages are reported.

Comment 12: I suggest deleting the rows „Total” from the tables.

Authors’ actions: Row total is not reported anymore, thank you for pointing, it was not useful.

Comment 13: I suggest including in the Discussion the information why breastfeeding rates aren’t high.

Authors’ actions: Thank you for your comment. We have not included in the analysis of the particular manuscript that information. This is planned to be part of a forthcoming manuscript looking in details these aspects related to breastfeeding.

Comment 14: I suggest including information concerning the strengths and limitations of the study.

Authors’ actions: Thank you for your advice. We have put a paragraph in the discussion section. The particular study had some important limitations. The major one appears to be that women in the intervention and control groups, participated voluntarily, thereafter might have had a predisposition to breastfeed, thus invoking the risk of bias, could be concealed. In addition, the participants of the study were recruited from only one healthcare unit/hospital in a large geographical region of Thessaly, and particularly in the city of Larissa, limiting in a way the safe establishment of clear outcomes in general population. The study did not include during postpartum period home visits but only counselling/training in the clinic and telephone breastfeeding support after discharge. Finally, it would be helpful to include also additional variables, such as husband’s and hospital training, etc. Although our study focused on the effectiveness of the midwifery intervention in this sample with positive results, future research would be useful to test the effectiveness of the midwifery intervention itself in larger populations.

Comment 15: Many bibliographies are a little I suggest updating the bibliography (2019 and 2020,2021) e.g.

Authors’ actions: We have updated the references. Thank you. We have added the following bibliography to the references section.

Souza EFDC, Pina-Oliveira AA, Shimo AKK. Effect of a breastfeeding educational intervention: a randomized controlled trial. Rev Lat Am Enfermagem. 2020 Sep 30;28:e3335. doi: 10.1590/1518-8345.3081.3335. eCollection 2020.

Yılmaz M, Aykut M.Review The effect of breastfeeding training on exclusive breastfeeding: a randomized controlled trial.J Matern Fetal Neonatal Med. 2021 Mar;34(6):925-932. doi: 10.1080/14767058.2019.1622672. Epub 2019 Jul 25.

Wang Y, You HX, Luo BR. Exploring the breastfeeding knowledge level and its influencing factors of pregnant women with gestational diabetes mellitus.

BMC Pregnancy Childbirth. 2020 Nov 23;20(1):723. doi: 10.1186/s12884-020-03430-9.

Li JY, Huang Y, Liu HQ, Xu J, Li L, Redding SR, Ouyang YQ. The Relationship of Previous Breastfeeding Experiences and Factors Affecting Breastfeeding Rates: A Follow-Up Study.Breastfeed Med. 2020 Dec;15(12):789-797. doi: 10.1089/bfm.2020.0165. Epub 2020 Sep 18.

Sehhatie FS, Mirghafourvand M, Havizari S. Effect of prenatal counseling on exclusive breastfeeding frequency and infant weight gain in mothers with previous unsuccessful breastfeeding: a randomized controlled clinical trial.J Matern Fetal Neonatal Med. 2020 Nov;33(21):3571-3578. doi:10.1080/14767058.2019.1579191. Epub 2019 Feb 17.

Kummer L, Duke N, Davis L, Borowsky I.Association of Social and Community Factors with U.S. Breastfeeding Outcomes.Breastfeed Med. 2020 Oct;15(10):646-654. doi: 10.1089/bfm.2020.0083. Epub 2020 Aug 26.

A systematic review of infant feeding food allergy prevention guidelines - can we AGREE? Vale SL, Lobb M, Netting MJ, Murray K, Clifford R, Campbell DE, Salter SM.

Breastfeeding promotion and support: a quality improvement study. Menichini D, Zambri F, Govoni L, Ricchi A, Infante R, Palmieri E, Galli MC, Molinazzi MT, Messina MP, Putignano A, Banchelli F, Colaceci S, Neri I, Giusti A.

Maternity ward staff perceptions of exclusive breastfeeding in Finnish maternity hospitals: A cross-sectional study.Hakala M, Kaakinen P, Kääriäinen M, Bloigu R, Hannula L, Elo S.

Increasing the exclusive breastfeeding rate in a private hospital in UAE through quality improvement initiatives. Kaushal M, Sasidharan K, Kaushal A, Augustine P, Alex M.

Thank you for revising our manuscript, we appreciate you time and comments.

Reviewer 2 Report

Dear Authors,

Thank you for allowing me to review your manuscript. I found it very interesting since it is necessary to increase research that demonstrates the effectiveness of breastfeeding support programs both to increase their rates in the population and thus improve their health status, as well as to demonstrate the important work that nurses and midwives perform with the aim of promoting the initiation and maintenance of breastfeeding. However, reading your manuscript raises some questions for me, especially from a methodological point of view, which I think it is important to correct or clarify.

Firstly, regarding the introduction, in my opinion it includes relevant and clear information, however, some points need to be revised. For example, you comment that “Evidence in the literature shows 74 that despite the fact that many women initiate breastfeeding, few finally meet the recom-75 mended goals in terms of duration and exclusivity”, you should cite the relevant studies and make reference to the origin of the sample of these studies.

On the other hand, it is not necessary to include the web addresses in the text ((https://www.healthypeople.gov/2020/topics-objectives/topic/maternal-infant-and-child-87 health/objectives); (https://www.who.int/health-top-88 ics/breastfeeding#tab=tab_1)), this information can be included in the references section.

In addition to the above, it would be useful to add in the introduction information on current literature on breastfeeding programmes in other countries that have been shown to be effective in increasing breastfeeding rates, as most of this information is only provided to discuss the results.

In my opinion, the methodology of the study has some shortcomings that should be rectified or revised. The first important aspect is to determine whether the population of low-risk women and women with endocrine disorders were randomly assigned to the Experimental Group or the Control Group. If not, the authors should clearly justify how and why the group allocation was made.

Another relevant aspect is to specify what type of intervention the women in the control group received, i.e. what the standard care consisted of.

On the other hand, The text indicates that the follow-up at 3 weeks, 4 months, 6 months, 1 year and 2 years was carried out by the midwife in charge; however, you do not explain the breastfeeding indicators measured at these follow-ups. It would be interesting to specify what these indicators are and explain them briefly in this parto f de text. Moreover, the authors do not indicate the duration of the study, when data collection started and ended.

Other details to be taken into account are to unify the use of the word breastfeeding because the word breast-feeding in not correct; to add the WHO reference on the definition of exclusive breastfeeding; to correct a typographical error in the sentence: “At 6 months after childbirth the 33.75% individuals of the midwifery intervention group with endocrine pregnancy disorder (33.755 vs 17.5% p=0.0019) continued to breastfeed…” and the "p" in p-value should be written in italics

With regard to the analyses carried out for the study, in my opinion it is important to make some improvements. It seems that, apart from the breastfeeding rates reported for each group at each evaluation time, the most complex analysis presented is the comparison between two groups using Chi-square, but the results are presented as correlations, which is an error. What type of correlation analysis between variables has been performed? Is the entire sample of women who received the intervention, i.e. low-risk women and women with endocrine disorders, considered for these comparative analyses? Furthermore, the OR values obtained are not reported and their interpretation is not adequate. I think the authors should reconsider these analyses and present them in a proper way again.

Regarding the results, I think that another relevant aspect to review is the format and content of all the tables presented:

In table 1, the sentences of the columns are too close together and it is confusing. Revise the content and summarise as it seems to be copied and pasted from the main text. In the variable "Nationality" the option "other" includes information on the percentage of Albanian women, it would be interesting to include the rest of the nationalities or to remove the Albanian option. Also, why is it important to include religion as a socio-demographic variable if no hypothesis is made about it in the study?

In the table 2 would be convenient to modify the distribution of the data, if you want to keep two columns, the answer options (i.e., married, divorced, single) should be in the characteristics column and the results in the values column.

The results reported in tables 3 to 7 are confusing. First of all, the text naming each table cannot contain that much information, it should be the title of the result in question. Moreover, it is not clear what the results of the row percentage and the column percentage contribute. In my opinion it would be easier to report n and % in each option and reorganise the information in one table instead of 7.

Table 8 requires a complete reorganisation in which the comparisons between variables and the OR calculations performed as well as the CIs can be clearly seen. In my opinion it is confusing.

Finally, I would like to comment that in my opinion, even though the programme works, it is not a programme that works, it is a programme that works. This study only reports rates but does not provide preliminary results that explain why the programme works or what conditions it might be related to. The design of the study could be improved, it needs more in-depth analysis to make a real contribution to the current literature.

I hope these suggestions can help you to improve your manuscript.

Author Response

List of changes

Reviewer 2

Comment 0:  Thank you for allowing me to review your manuscript. I found it very interesting since it is necessary to increase research that demonstrates the effectiveness of breastfeeding support programs both to increase their rates in the population and thus improve their health status, as well as to demonstrate the important work that nurses and midwives perform with the aim of promoting the initiation and maintenance of breastfeeding. However, reading your manuscript raises some questions for me, especially from a methodological point of view, which I think it is important to correct or clarify.

Authors’ actions: Thank you. We included in table 1.2 Standard Hospital Intervention to clarify the difference to the specialist midwife intervention.

Comment 1:  Firstly, regarding the introduction, in my opinion it includes relevant and clear information, however, some points need to be revised. For example, you comment that “Evidence in the literature shows 74 that despite the fact that many women initiate breastfeeding, few finally meet the recom-75 mended goals in terms of duration and exclusivity”, you should cite the relevant studies and make reference to the origin of the sample of these studies.

Authors’ actions: Relative references have been added.

Comment 2: On the other hand, it is not necessary to include the web addresses in the text ((https://www.healthypeople.gov/2020/topics-objectives/topic/maternal-infant-and-child-87 health/objectives); (https://www.who.int/health-topics/breastfeeding#tab=tab_1)), this information can be included in the references section.

Authors’ actions: Thank you for your comment. We have put webpage to the references.

Comment 3: In addition to the above, it would be useful to add in the introduction information on current literature on breastfeeding programmes in other countries that have been shown to be effective in increasing breastfeeding rates, as most of this information is only provided to discuss the results.

Authors’ actions: Thank you for your remark. Similar breastfeeding programmes like our specialist midwife interventions that are characterized as effective for initiation and the duration of breastfeeding were introduced in other countries and have been shown to be effective in increasing breastfeeding rates and the relative references have been addressed in the introduction.

Comment 4: In my opinion, the methodology of the study has some shortcomings that should be rectified or revised. The first important aspect is to determine whether the population of low-risk women and women with endocrine disorders were randomly assigned to the Experimental Group or the Control Group. If not, the authors should clearly justify how and why the group allocation was made.

Authors’ actions: In the revised version, section “2.1. Study population” it is clarified that the women were randomly assigned to the control and intervention groups.

Comment 5: Another relevant aspect is to specify what type of intervention the women in the control group received, i.e. what the standard care consisted of.

AuthorsactionsThank you for your remark. Please find below the requested information. We have also added Table1.2. in the revised version of our manuscript.

MIDWIFERY INTERVENTION    

PRENATAL                                                                                                          

  • Personalized prenatal midwifery program (Structured midwifery education-support, after 34W of gestation

POSTNATAL

  • Personalized postnatal structured midwifery program. Structured midwifery education and support (the position and attitude of mother holding baby, method of supporting breast infant’s latching on to breast when suckling, sings of effective suckling, the resolution of breastfeeding problems and direct support the recognition of signs of nutritional adequacy of the newborn)(one visit- postpartum at 2rd-3rd day)

  • Telephone communication from the midwife, to empower women, support them for any difficulties of breast-feeding and provide information (Midwifery Telephone support and counselling, it was possible for participants to contact the intervention midwife in charge for any questions beyond the scheduled telephone communication)

STANDARD CARE

PRENATAL

  • Standard care doesn’t include any prenatal program for breastfeeding
  • Written breastfeeding policy unfortunately does not exist, to be routinely communicated to all health care staff

POSTNATAL

  • The midwife help mothers initiate breastfeeding within 2 hours after birth.
  • The health care staff educates and supports only mothers who want to breastfeed
  • Designated breastfeeding and breast milk expression area
  • Rooming in
  • Lactation consultants and other breastfeeding experts on patient care teams does not exist
  • Standard care doesn’t include any telephone communication from midwife at 3 weeks, 4 months, 6 months, first and second year after birth

Comment 6: On the other hand, the text indicates that the follow-up at 3 weeks, 4 months, 6 months, 1 year and 2 years was carried out by the midwife in charge; however, you do not explain the breastfeeding indicators measured at these follow-ups. It would be interesting to specify what these indicators are and explain them briefly in this parto f de text. Moreover, the authors do not indicate the duration of the study, when data collection started and ended.

Authors’ actions: The study period was from March 2017 to March 2020. This was added to the revised version of our manuscript. Thank you.

Comment 7: Other details to be taken into account are to unify the use of the word breastfeeding because the word breast-feeding in not correct; to add the WHO reference on the definition of exclusive breastfeeding; to correct a typographical error in the sentence: “At 6 months after childbirth the 33.75% individuals of the midwifery intervention group with endocrine pregnancy disorder (33.755 vs 17.5% p=0.0019) continued to breastfeed…” and the "p" in p-value should be written in italics

Authors’ actions: Thank you for your comment. We presented our data according to the journal guidelines. We hope that minor changes could be corrected during typesetting.

Comment 8: With regard to the analyses carried out for the study, in my opinion it is important to make some improvements. It seems that, apart from the breastfeeding rates reported for each group at each evaluation time, the most complex analysis presented is the comparison between two groups using Chi-square, but the results are presented as correlations, which is an error. What type of correlation analysis between variables has been performed? Is the entire sample of women who received the intervention, i.e. low-risk women and women with endocrine disorders, considered for these comparative analyses? Furthermore, the OR values obtained are not reported and their interpretation is not adequate. I think the authors should reconsider these analyses and present them in a proper way again.

Authors’ actions: Thank you for your remark. Actually, there is no correlation analysis! The word correlation was used by mistake, in the revised version this is expressed using different terms (please see revised document with tracked changes). After combining tables 3-7 into table 3 (in the revised version) we have left only the row percentages (as suggested by the reviewers), thus we have modified the statistical analysis in lines 299-330 to report in the ORs and the relevant 95% CIs.

Comment 9: Regarding the results, I think that another relevant aspect to review is the format and content of all the tables presented:

In table 1, the sentences of the columns are too close together and it is confusing. Revise the content and summarise as it seems to be copied and pasted from the main text. In the variable "Nationality" the option "other" includes information on the percentage of Albanian women, it would be interesting to include the rest of the nationalities or to remove the Albanian option. Also, why is it important to include religion as a socio-demographic variable if no hypothesis is made about it in the study?

Authors’ actions: Thank you for your comment. Table 1 has been updated.

Comment 10: In the table 2 would be convenient to modify the distribution of the data, if you want to keep two columns, the answer options (i.e., married, divorced, single) should be in the characteristics column and the results in the values column.

Authors’ actions: Table 2 is now modified as suggested.

Comment 11: The results reported in tables 3 to 7 are confusing. First of all, the text naming each table cannot contain that much information, it should be the title of the result in question. Moreover, it is not clear what the results of the row percentage and the column percentage contribute. In my opinion it would be easier to report n and % in each option and reorganise the information in one table instead of 7.

Authors’ actions: Tables 3-7 were reported also confusing by another reviewer, these tables are now unified and less information is presented.

Comment 12: Table 8 requires a complete reorganisation in which the comparisons between variables and the OR calculations performed as well as the CIs can be clearly seen. In my opinion it is confusing.

Authors’ actions: Table 8 (in the revised version is table 4) is now re-organized, hope it is easier to understand, however it is not possible to have ORs and the relevant CIs because there are more than two categories in the variables (specifically it is about breastfeeding for: a) First days b) ≤ 4 months and c) ≤ 6 months)

Comment 13: Finally, I would like to comment that in my opinion, even though the programme works, it is not a programme that works, it is a programme that works. This study only reports rates but does not provide preliminary results that explain why the programme works or what conditions it might be related to. The design of the study could be improved, it needs more in-depth analysis to make a real contribution to the current literature.

Authors’ actions: Thank you for revising our manuscript, we appreciate you time and comments.

Round 2

Reviewer 1 Report

You have done a great job. I suggest including in the limitations of the study the information that patients with diabetes and thyroid diseases should be in the separate groups and it should be analysed in the next experiment.

Reviewer 2 Report

Many thanks to the authors for their efforts to improve the study. In general, the changes made respond to most of the comments. Several aspects of the methodology and analyses needed to be reviewed in detail. One important aspect of the methodology is that the sample was randomised in the distribution of the groups and this is now clear. However, although improvements can be seen in 3.2.1 and table 3 has been added, it remains unclear how the comparison of groups has been done. The authors explain that they compare the intervention and control groups by OR but this is not justified. The analyses are shallow and remain unclear in their wording. In my opinion this point should be improved to enhance the purpose of the study. In this sense, to extend the analyses and not only compare breastfeeding rates but also explore the differences between the two groups of women perhaps the following information would help: "To explore the differences between women who had received the intervention program and those who did not, the authors can computed a Chi-square analysis for the dichotomous variables and a Mann-Whitney "U" analysis or Kruskal Wallis "H" for the quantitative variables if the study variables did not follow a normal distribution. The odds ratio (OR) could be calculated when significant differences were found between variables".
If they finally decide to only present the results regarding the improvement of breastfeeding rates, it would be convenient to revise the wording regarding ORs (e.g., The results showed that the women who had received midwifery intervention were XX-fold (OR=XX [95%CI: XXX-XXX] ) more likely to continued in brestfeeding (χ² (XXX, p<.XX)).
These suggestions are applicable for point 3.3 where the authors speak of relationship (formerly correlation) and the analyses are too simple for this.
Finally, the changes made should be reflected in the discussion.

Despite these comments, in my opinion, it is worth making the effort to improve since your study is very valuable and shows that breastfeeding support programmes can be effective.